# Modulation of serotonin transporter expression by escitalopram under inflammation
Sergio Mena[1], Allison Cruikshank [2], Janet Best[3], H. F. Nijhout[4], Michael C. Reed[2] & Parastoo Hashemi [1]✉

Selective serotonin reuptake inhibitors (SSRIs) are widely used for depression based on the monoamine deficiency hypothesis. However, the clinical use of these agents is controversial, in part because of their variable clinical efficacy and in part because of their delayed onset of action. Because of the complexities involved in replicating human disease and clinical dosing in animal models, the scientific community has not reached a consensus on the reasons for these phenomena. In this work, we create a theoretical hippocampal model incorporating escitalopram's pharmacokinetics, pharmacodynamics (competitive and non-competitive inhibition, and serotonin transporter (SERT) internalization), inflammation, and receptor dynamics. With this model, we simulate chronic oral escitalopram in mice showing that days to weeks are needed for serotonin levels to reach steady-state. We show escitalopram's chemical efficacy is diminished under inflammation. Our model thus offers mechanisms for how chronic escitalopram affects brain serotonin, emphasizing the importance of optimized dose and time for future antidepressant discoveries.

Depression is the leading cause of global disease burden[1]. The condition is known to increase absenteeism at work, the risk of suicidal tendencies, and other conditions such as cardiovascular disease[2,3]. The most prescribed antidepressants, selective serotonin reuptake inhibitors (SSRIs), directly target the serotonin system, more specifically the serotonin transporter (SERT), on a rationale built upon the monoamine hypothesis of depression[4]. Despite a substantial body of evidence from animal and clinical studies supporting the hypothesis[5–10] and the potential efficacy of SSRIs[11], their clinical use remains a subject of controversy. Two key contributing factors to this controversy include the variable clinical efficacy (30-60%, depending on the study)[12–14] and the extended period of chronic administration required for clinical relief[15]. Due to the challenges faced when modeling these two issues in animals, the scientific community has not yet reached a consensus on the underlying reasons for these phenomena. Consequently, there is increasing interest in exploring alternative treatments, such as ketamine and psychedelics[16–20], that offer mechanisms of action beyond the monoamine hypothesis.

Our group has been at the forefront of developing and utilizing real-time fast-scan cyclic voltammetry (FSCV) tools for serotonin detection in animal models. Our recent findings have revealed serotonin as a promising biomarker for stress-induced depression phenotypes in mice (as a

consequence of inflammatory signaling processes). Further, we found that escitalopram was chemically less able to increase serotonin during inflammation[6,21]. Finally, we confirmed that serotonin is a common target linking various classes of antidepressants[22]. Collectively, our studies provide compelling evidence for the important role of serotonin in the pathology of depression, but also underscore the need for a more nuanced understanding of how this messenger is modulated during depression and chronic SSRI dosing.

Given the difficulty of modeling these complex clinical phenomena experimentally, here we adopted a theoretical approach based on our prior experimental data in the hippocampus. We chose to focus on escitalopram, which is thought to be one of the most clinically efficacious SSRIs due to its allosteric binding to SERTs and rapid induction of SERT internalization[22–25] and the known interaction of this agent in the hippocampus[26].

First, we developed a model, comprising 51-differential equations, that incorporated orthosteric and allosteric binding of escitalopram to SERT, SERT internalization, inflammation, and receptor dynamics to fit our experimental FSCV data under acute escitalopram. Second, we simulated chronic oral escitalopram dosing in mice, analogous to human clinical treatment, and found that extracellular serotonin needs significant time to reach a new steady-state after the onset of oral dosing, as a function of

¹Department of Bioengineering, Imperial College London, London SW7 2AZ, UK. ²Department of Mathematics, Duke University, Durham, NC, USA. ³Department of Mathematics, The Ohio State University, Columbus, OH, USA. ⁴Department of Biology, Duke University, Durham, NC, USA. ✉e-mail: phashemi@imperial.ac.uk

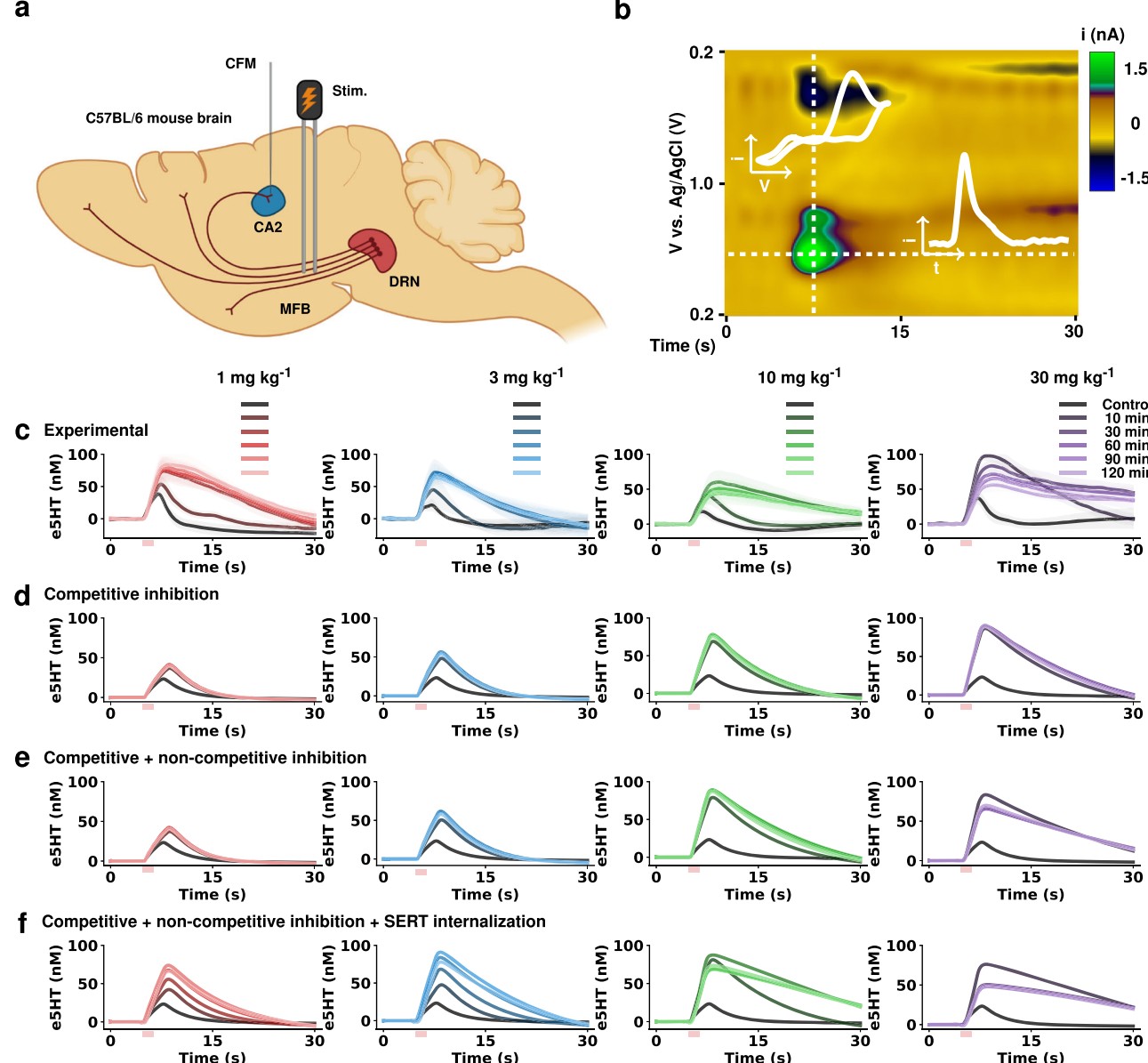

**Fig. 1 | Modeling evoked FSCV antidepressant data. a** Schematic of experimental protocol for evoked FSCV serotonin signals in the CA2 region of the hippocampus. **b** Representative FSCV color plot control file. Inset: The vertical trace (i vs. v) shows a representative serotonin CV, while the horizontal trace (i vs. t) shows the release and reuptake profile of serotonin over the course of the experiment. **c** Experimental concentration vs. time traces from the CA2 region of the hippocampus of female mice ($n = 4$ animals each dose, mean ± SEM) pre- and post-escitalopram intraperitoneal administration at time and dose given in the legend. **d** Simulation of serotonin extracellular concentration pre- and post-escitalopram at time and dose given in the legend solely assuming competitive inhibition. **e** Simulation following the same protocol as in D, adding non-competitive inhibition. **f** Simulation following the same protocol as in E and adding SERT internalization. Illustrations made with Biorender.com.

autoreceptors. Finally, we simulated inflammation *via* histamine released from mast cells and glia and demonstrated how inflammation-induced histamine inhibits serotonin via $H_3$ receptors, offering a possible reason for why under inflammation, SSRIs may be less effective.

Our study proposes potential mechanisms that underlie the important facets of escitalopram and stresses the importance of considering the optimal dose and timing of therapy when designing future antidepressant drugs.

## Results

### Allosteric binding and rapid SERT internalization are required to model acute escitalopram's effect on serotonin

In this study, we utilized previously published experimental data where four different acute doses of escitalopram were used in mice[22]. In the experiment,

cohorts of mice were anesthetized, and serotonin release was stimulated in the CA2 region of the hippocampus by electrical stimulation of the medial forebrain bundle (MFB) (Fig. 1a). To measure the release and reuptake of serotonin, we performed FSCV. An example of raw data, "a color plot", of the response upon this stimulation is in Fig. 1b. Figure 1c shows experimental measurements of evoked serotonin release prior to and at various time points post-drug administration with four different escitalopram doses.

We previously utilized a one-differential equation model for experimental data in mice after antidepressants[27]. This simple model failed to successfully explain the dynamic dose-response data shown in Fig. 1c. Primarily, we were not able to model the strong decrease of reuptake rate in all doses and the decrease in amplitude at higher doses (10 and 30 mg kg$^{-1}$).

Therefore, we combined previously developed models that include several additional biochemical processes relevant to serotonin and histamine neurotransmission, including synthesis, metabolism, release and reuptake in terminals and glia[28,29], and a model of the pharmacokinetics (PK) of escitalopram after intraperitoneal (i.p.) injection[22].

To represent this data, we had to incrementally refine the combination of models to include escitalopram-specific phenomena. First, we simulated the experimental results by modeling only escitalopram's competitive inhibition of SERTs (Fig. 1d) and found that we could not capture the data well. As demonstrated by the simulated traces, the decrease in reuptake rate is not as observed in the experimental data. Specifically, for lower doses, the amplitude of the evoked trace does not increase proportionally, as seen in the post-drug experimental traces, and the reuptake speed is much faster. To quantify the disparity between experimental and simulated data, we first calculated the maximum amplitude of the trace and the half-life of evoked serotonin for experimental and simulation traces (see Fig. S1 in Supplementary Information). Additionally, we calculated the root mean squared error (RMSE) and mutual information score (MI) between experimental and simulated traces. The average error over all traces (dose and time) was RMSE = 19.21 ± 1.12 nM and agreement of information MI = 0.66 ± 0.03. We sought to simultaneously decrease the error, increase the mutual information between simulations and experimental, and visually maximize the similarity between simulated and experimental data.

Next, we added escitalopram's non-competitive inhibition of SERTs to the model (Fig. 1e)[23]. The simulated traces have an increase in evoked amplitude and, as escitalopram dose increases, the reuptake strength decreases in comparison to Fig. 1d. Additionally, the agreement between experimental and simulation traces increased (RMSE = 16.06 ± 1.26 nM, MI = 0.84 ± 0.04). While these simulations are closer to experimental data, they visually still do not fully capture the data. Finally, by adding rapid SERT internalization[22] to the model, we were able to capture the data most closely (Fig. 1f), as well as further reducing the deviation from experimental (RMSE = 15.23 ± 1.08 nM, MI = 1.01 ± 0.05). These changes in SERT density are dynamic and occur directly and rapidly instantaneously as a function of escitalopram concentration. The modeled traces for lower doses (1 and 3 mg kg$^{-1}$) at time 30 min are not as pronounced as the experimental data. The reuptake at higher doses (10 and 30 mg kg$^{-1}$) at time 10 min post-administration is slower than in the experimental data, implying that there may be further mechanisms not currently included in the model, responsible for these responses.

These simulations are much closer to the experimental data, since they show the substantial decrease in reuptake rate evidenced by the greatly decreased slope of reuptake. Additionally, measurements of amplitude and clearance rate (see Fig. S1 in Supplementary Information) show a much clearer agreement between experimental data and the simulations where we modeled the three escitalopram effects.

Therefore, to model dynamic dose-response data, we combined several previously models, and added escitalopram's competitive inhibition[30], non-competitive inhibition[23,24] and SERT internalization[22,31,32]. The model that fits these data comprises 51 differential equations (see Supplementary Information for full mathematical description of the model). The next section presents this new model.

## The model of escitalopram-modulated serotonin
Figure 2 displays the schematic of the main processes incorporated in the computational model. Table 1 provides descriptions of each variable depicted in the model, including the effects induced by drugs. Table 2 describes each enzyme, transporter, and receptor.

The computational model encompasses all relevant processes, considered by us, involved in the synthesis, release, and reuptake of serotonin, in synaptic terminals (Fig. 2a). The model includes the well-described (by us and others) inhibition of serotonin release by histamine $H_3$ receptors[33–35]. Additionally, the model includes a comprehensive representation of the ability of glia (microglia and astrocytes) to reuptake both histamine and serotonin and, in some cases, synthesize histamine (Fig. 2b). Finally, we

introduce a novel representation of the dynamic effects of escitalopram on serotonin reuptake dynamics (Fig. 2c). The escitalopram modeling began progressively with the creation of a four-compartment PK model of escitalopram after an i.p. injection and modeling the competitive effects of the drug on serotonin reuptake using a single equation. We subsequently added non-competitive inhibition (effects on the inhibition constant of escitalopram), as well as the influence of both serotonin receptors and escitalopram on SERT internalization (SERT pool trafficking). A detailed description of the model equations, variables, and functions, along with a definition of the equilibrium state, can be found in the Supplementary Information.

The ability of this model to closely represent and simulate important processes, such as autoreceptor control of cytosolic processes and the availability of a readily releasable pool of vesicles, allows for a closer representation of the effects of escitalopram observed in the experimental data (Fig. 1c). We thus use this model to study phenomena that are difficult to experimentally test.

## Mathematically modeling chronic escitalopram administration
Clinically, escitalopram and other SSRIs are taken orally on a daily basis. The dose regime is designed so that consistent levels of the drug remain in the plasma; commonly, the dose is repeated before the elimination half-life, which for humans, is around 27–32 h[36].

To model this chronic oral regime in mice in silico, we had to modify our model, which was initially designed to represent acute, intraperitoneal administration in mice. First, we decreased the adsorption rate and bioavailability of the drug to represent the adsorption differences between the gut and the peritoneum (see methods). Figure 3ai depicts the simulated time course of escitalopram in the brain for an i.p. vs. oral administration. The decreased amplitude but increased duration of the oral trace is a result of the lower adsorption rate; escitalopram is adsorbed slower when taken orally, so the plasma and brain concentrations reached are lower, and the drug takes longer to clear[37]. In rodents, due to their faster rate of metabolism compared to humans, the i.p. elimination half-life is much lower[38,39], and so is our estimated oral elimination half-life (~10 h). Taking these parameters into account, we simulated a chronic paradigm by repeating the administered dose every 8 h. Figure 3aii and iii show a representative brain escitalopram simulation where dosing occurs every 8 h in the hours range and days range, showing how the overall concentration of escitalopram builds up in the system over time in a mouse.

We next simulated the system response to an oral administration and compared it to the previous intraperitoneal results. Figure 3b shows simulations of *i.p.* and oral administration of a mouse-equivalent dose of 5 mg escitalopram (~1.02 mg kg$^{-1}$, see methods for conversion procedure). The decrease in amplitude and slower elimination half-life of brain escitalopram (i) are transferred to the effects of escitalopram on extracellular serotonin (ii), and SERT density (iii). Then, repetition of the oral administration was simulated every 8 h for 15 days. Figure 3c shows the response of brain escitalopram (i), extracellular serotonin (ii), and SERT density (iii) for the chronic oral paradigm. As mentioned earlier, escitalopram reaches a steady-state concentration of ~1–2 after the first dose, while serotonin and SERT density overshoot, not reaching stability for 7–8 days after the first dose.

One reason the model puts forth is the autoreceptors finding a new steady-state based on the increasing levels of serotonin after escitalopram. In the model, the mechanism behind the overshoot is the temporal difference between the effects of escitalopram on serotonin levels (fast) and the slower effect of autoreceptor regulation of serotonin via decreased synthesis and release. Escitalopram rapidly slows down the reuptake of serotonin, near instantly increasing extracellular serotonin and those higher levels of serotonin activate the autoreceptors, which inhibit the synthesis and firing of serotonin and decrease extracellular serotonin. In the model, one equation captures the concentration of serotonin after escitalopram while a chain of equations describes the effects of the autoreceptors, meaning that early on in the chronic dosing regime, the effects of escitalopram on serotonin outrun the effects of autoreceptors, hence the overshoot. Over some days, the

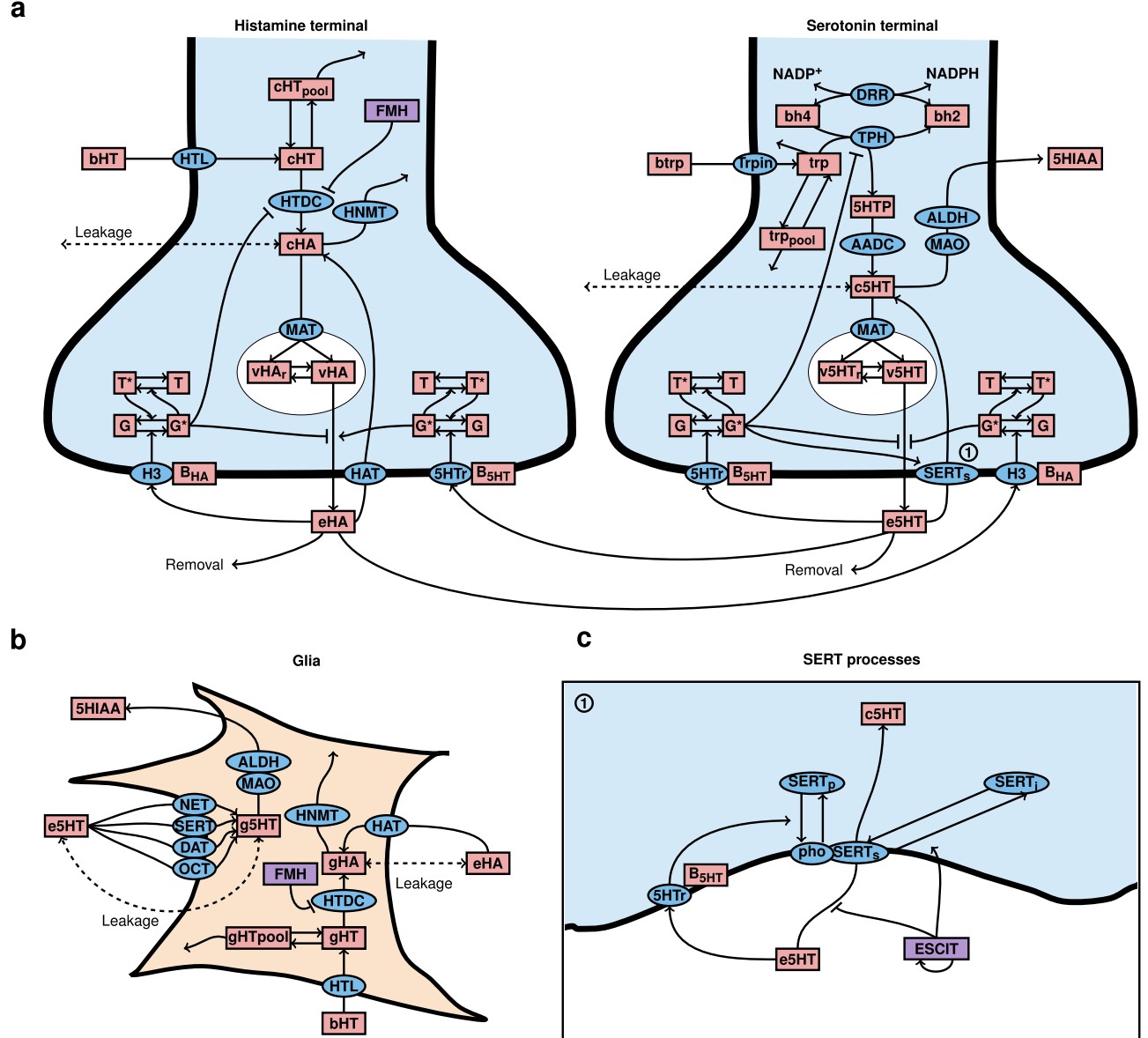

**Fig. 2 | Representation of the relationships in the mathematical model of serotonin, histamine, and glia. a** Schematic of serotonin and histamine neuronal terminals. **b** Schematic of glia. **c** Detailed modeled SERT processes. Symbols of the model's main variables are depicted in red rectangular labels and described in Table 1. The acronyms of main enzymes, transporters, and receptors are represented with blue elliptic labels and described in Table 2. The pharmacological effects of escitalopram

(ESCIT) and α-fluoromethylhistidine (FMH) are represented in purple rectangular labels. Notice that variables and enzymes with the same name in the schematic but present in different compartments are different entities (e.g., HTDC in histamine neurons and HTDC in glia). An extended description of each variable and enzyme/transporter by compartments can be found in the Supplementary Information. Illustration made with LaTeX Tikz pachage.

autoreceptor effect catches up and a new steady-state is reached. As the dose increases, the new steady-state concentration is higher, and it is reached faster (Figs. S2, S3).

Having shown that serotonin levels take considerable time to reach a steady-state after chronic oral dosing with escitalopram in a control situation due to autoreceptors seeking a new state, we next asked how these effects are mediated under an inflammation model of depression.

**Chronic escitalopram during inflammation**

To simulate an inflammation model of depression, we drew on previous experimental work that showed increased brain histamine in mice that underwent a chronic stress paradigm and thus developed behavioral phenotypes of depression[6]. We simulated this increase in histamine via glial

activation and mast cell degranulation (Figs. 2b, 4ai). Figure 4ai shows the biochemical mechanisms that modulate histamine release from mast cells. Figure 4aii depicts a proposed model of how extracellular histamine levels in the hypothalamus change in response to inflammation and the subsequent effects on serotonin (histamine inhibits serotonin via H₃ heteroreceptors on serotonin neurons). Figure 4b is analogous to Fig. 3c, showing how chronic escitalopram performs under inflammation.

To better compare the difference between control and inflammation, we present the two data sets in Fig. 4bii. The control starting ambient levels are 60.0 nM, compared to values under inflammation, 47.5 nM. After chronic escitalopram in control the serotonin levels stabilize with a mean of 69.78 nM (16% increase) and under inflammation this is 53.62 nM (12% increase)—it is worth noting that chronic escitalopram under inflammation

**Table 1 | Description of main model variables of compartmental concentrations**

| Name | Description |
|---|---|
| bHT | Blood histidine. |
| cHT | Cytosolic histidine. |
| $cHT_{pool}$ | Cytosolic histidine pool. |
| cHA | Cytosolic histamine. |
| $vHA_r$ | Reserve of vesicular histamine. |
| vHA | Vesicular histamine. |
| $B_{HA}$ | Bound histamine to receptors. |
| $B_{5HT}$ | Bound serotonin to receptors. |
| eHA | Extracellular histamine. |
| T | Regulator of G protein in receptors. |
| T* | Activated regulator of G protein in receptors. |
| G | G protein in receptors. |
| G* | Activated G protein in receptors. |
| btrp | Blood tryptophan. |
| trp | Cytosolic tryptophan. |
| $trp_{pool}$ | Cytosolic tryptophan pool. |
| bh4 | Tetrahydrobiopterin. |
| bh2 | Dihydrobiopterin. |
| $NADP^+$ | Nicotinamide adenine dinucleotide phosphate. |
| NADPH | Reduced nicotinamide adenine dinucleotide phosphate. |
| 5HTP | 5-hydroxytryptophan or oxitriptan. |
| c5HT | Cytosolic 5-hydroxytryptamine or serotonin. |
| 5HIAA | 5-Hydroxyindoleacetic acid |
| $v5HT_r$ | Reserve of vesicular serotonin. |
| v5HT | Vesicular serotonin. |
| e5HT | Extracellular serotonin. |
| g5HT | Glial cytosolic serotonin. |
| gHT | Glial histidine. |
| $gHT_{pool}$ | Glial histidine pool. |
| gHA | Glial histamine. |
| ESCIT | Escitalopram. |
| FMH | (S) α-fluoromethylhistidine. |

**Table 2 | Description of main enzymes, transporters, receptors, and others**

| Name | Description |
|---|---|
| HTL | Histidine transporter. |
| HTDC | Histidine decarboxylase. |
| HNMT | Histamine methyltransferase. |
| MAT | Vesicular monoamine transporter. |
| HAT | Histamine transporter. |
| H3 | Histamine $H_3$ receptor. |
| $5HT_r$ | Serotonin receptor (presumably 5-$HT_{1B}$ receptor). |
| Trpin | Neutral amino acid transporter. |
| DRR | Dihydrobiopterin reductase. |
| TPH | Tryptophan hydroxylase. |
| AADC | Aromatic amino acid decarboxylase. |
| MAO | Monoamine oxidase. |
| ALDH | Aldehyde dehydrogenase. |
| $SERT_s$ | Serotonin transporter in the membrane surface. |
| pho | Subsection of phosphorylated serotonin transporters in the surface. |
| $SERT_p$ | Serotonin transporter in the pool. |
| $SERT_i$ | Inactivated serotonin transporter. |
| NET | Norepinephrine transporters. |
| DAT | Dopamine transporter. |
| OCT | Organic cation transporters. |
| Leakage | Diffusion through cell membrane. |
| Removal | Removal from the model (e.g., diffusion out of the system, capillary reuptake, etc.) |

is not able to restore serotonin to control, baseline levels. Figure 4biii shows how SERT density tracks the serotonin concentrations under control and inflammation. Motivated to target this inflammation, we modeled the inhibition of histamine synthesis. Figure 4ci is a simulation of how an acute i.p dose of a histamine synthesis inhibitor (alpha fluoromethylhistidine (FMH)) changes histamine over 25 h and the subsequent change in serotonin in Fig. 4cii). Figure 4di simulates the inflammation response after chronic SSRI and acute FMH. Figure 4dii and iii show a superimposition of the control response to chronic SSRI (as in Fig. 3c) and now chronic SSRI and acute FMH under inflammation. This pharmacological regime mirrors the control rather well.

## Discussion

The community has come to appreciate that a simple reuptake model, based on competitive inhibition via a change in SERT affinity, does not fully capture the much more complex mechanism of action of SSRIs[31,40,41]. We have very much mirrored this realization in our work, which can be illustrated here in real time using Fig. 1. Here we took experimental data, averaged between cohorts of mice, that received different acute doses of escitalopram over 2 h. In the past, we mathematically modeled such profiles with a 1-differential equation model that expressed the rate of change of

serotonin as a product of only three processes (release, Uptake 1 and 2, and autoreceptors)[27,42]. Escitalopram competitive inhibition was simulated by decreasing the affinity of transporters for serotonin ($K_m$). This simple model failed to fit the strong decrease of reuptake rate for all doses and the decrease in amplitude at higher doses (10 mg kg$^{-1}$ and 30 mg kg$^{-1}$)[22] in experimental data. Over time, we iterated the model[43] and here, to fit the experimental data in Fig. 1, we added a very high level of complexity; 51 differential equations.

This complex model includes serotonin synthesis, release, reuptake, and metabolism in terminals and glia. Auto- and hetero- modulation of release and regulation of synthesis and reuptake by autoreceptors and pharmacokinetics and pharmacodynamics (PK/PD) of escitalopram and FMH. Notably, there are two important facets of the model that we'd like to highlight, that were critical to fitting this escitalopram data.

First is the addition of allosteric modulation of SERTs by escitalopram. The allosteric site of SERTs is a secondary binding site which modulates the function of the transporter while not competing with serotonin binding[44]. In recent years, it is increasingly thought that escitalopram's effectiveness is dependent on this allosteric binding[23,30,45,46]. Although the mechanism of action is not clear, escitalopram binding to the SERT allosteric site is known to decrease the dissociation constant of escitalopram binding to the orthosteric site and prolong its inhibitory effects[47,48].

Second is escitalopram's ability to downregulate SERT membrane density. This phenomenon has been recently under the spotlight[25,41,49–51] and while the mechanisms are not fully understood, they are thought to be independent of allosteric binding[51]. The internalization process has been reported to occur over hours (cell culture) and days (humans)[32,50]. We recently showed in human-derived serotoninergic neurons that escitalopram can induce this process very rapidly (minutes)[22], and indeed, our model required this process to be fast.

Thus, we developed a complex model, comprising 51 differential equations, that fits a temporo-dose-response of acute escitalopram in vivo in

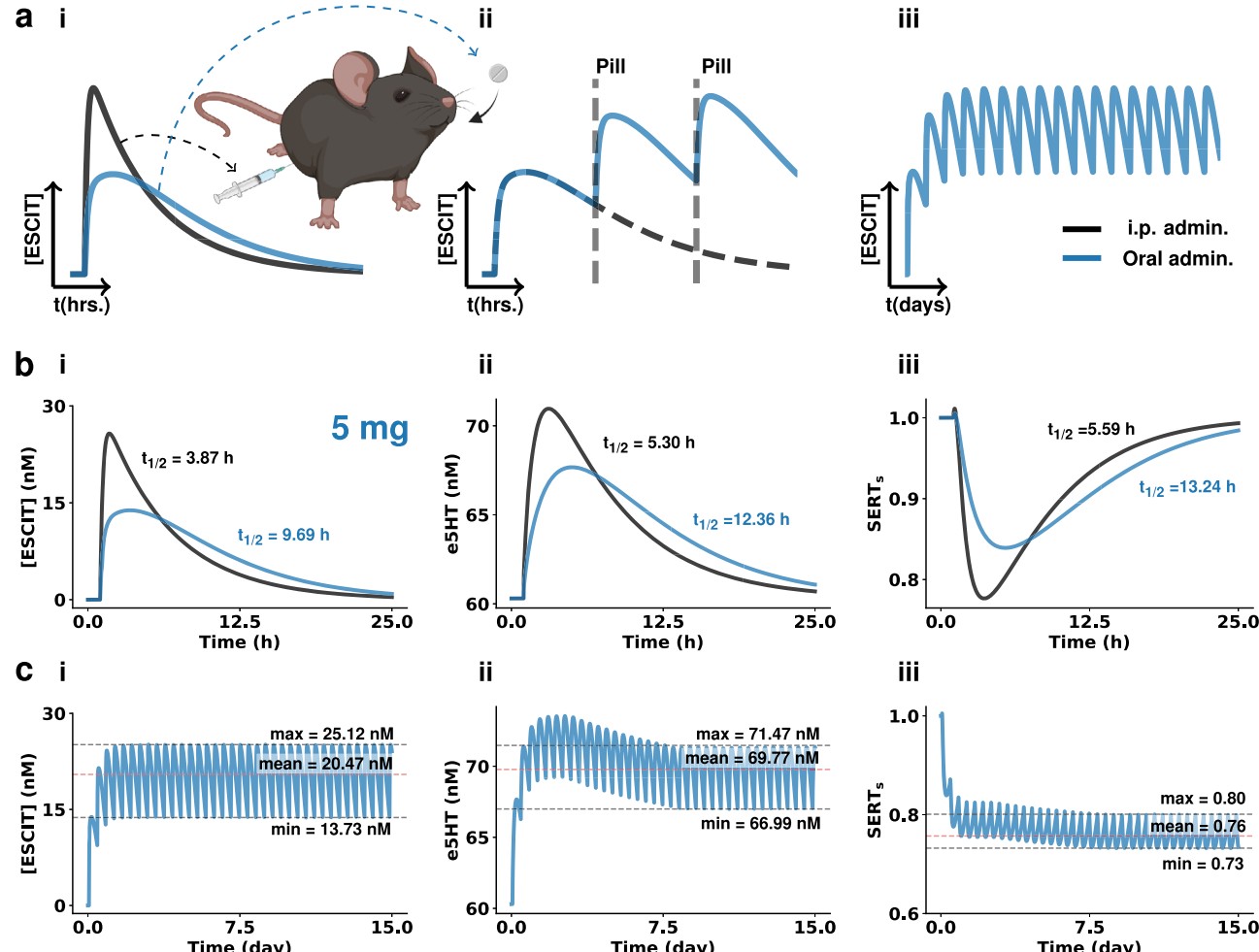

**Fig. 3 | Model of oral chronic dosing with escitalopram. a** Oral dosing paradigm, (i) shows the differences in escitalopram time profiles in the brain after i.p. and oral administration, (ii) shows the increase of escitalopram when the administration is repeated before clearance, and (iii) shows how escitalopram reaches a steady-state after several administrations. **b** Modeling of brain concentration of escitalopram (i), serotonin (ii), and SERT surface ratio (iii) following an i.p. injection or oral administration of 1.04 mg kg$^{-1}$ (equivalent to 5 mg human dose, see Methods). Half-

lives of the clearance are provided in the panels. **c** Modeling oral chronic dosing effects on escitalopram (i), serotonin (ii), and SERT surface ratio (iii). Administration is repeated every 8 h (equivalent to human daily dose, see Methods section). Maximum and minimum values of oscillations, as well as the mean accumulated concentration is given in the panels. Results for the 10 mg and 20 mg human doses can be found in the Supplementary Information Information (Figs. S2, S3). Illustrations made with Biorender.com.

the hippocampus, critically as a function of allosteric binding and rapid SERT internalization. It is important to note that this model may not extend to other brain regions, as there are regional differences in reported SSRI effects[52–54]. We next applied this model in a bid to provide an explanation for why, clinically, escitalopram can take several weeks to have a therapeutic effect.

Behavioral studies of depression are complicated and controversial in rodent models[55,56] and the community has recently become concerned about the validity of commonly used behavioral tests to accurately reflect depression phenotypes[57–60]. The forced swim test (FST)[61] is a famous example and has been used extensively (including by us)[6,62] in rodents as an index of depressive behavior. A relatively robust finding using the FST is that the length of time it takes for non-depressed animals to enter a learned helplessness state can be improved by acute injections of antidepressants[63]. However, few rodent studies have appropriately captured such behavioral changes in response to chronic antidepressants in depressed animals[64–66]. In addition, while many agents (given acutely to non-depressed animals) created this behavioral shift in rodents, they failed in humans[67,68]. Where antidepressants do show clinical efficacy in humans, the onset of action usually takes several weeks of chronic administration (once-a-day oral pill)[69,70]. The reason for this delay is unknown, and is very difficult (if not

impossible) to model in animals. Animals will not willingly swallow a pill, and other experimental ways to administer a chronic dose fall short in reproducing the pharmacokinetics of a human dose regime. This is because in humans, the concentration of the active agent oscillates around a period between a single pill administration, dependent on the pharmacokinetics of the agent and the characteristics of agent release from the pills[71]. This is also true for escitalopram.

Thus, in the absence of an appropriate experimental model, we found an opportunity here to apply our computational model to the investigation of oral dosing of escitalopram. Using the model described above, we changed the PK model of escitalopram to represent oral chronic dosing. We did this by decreasing the absorption rate of escitalopram (to represent the lower rate of absorption in the digestive tract) and obtained the FDA-recommended equivalent doses in mice of the most prescribed Lexapro™ tablets (5, 10, and 20 mg)[72]. Additionally, we repeated the dose every 8 h (as opposed to 24 h) to account for the faster metabolism of the drug in mice ($t_{1/2}^{mice}$ = 9,69 h vs. $t_{1/2}^{human}$ = 27-33 h)[36,37]. The key idea underlying oral chronic dosing theory is that the dose regime should build a desired steady-state concentration of the drug in plasma over time. To build such a steady-state, a new dose is taken before the previous dose has been fully metabolized[73], guided by the $t_{1/2}$ of the agent (see Methods section). When we modeled this

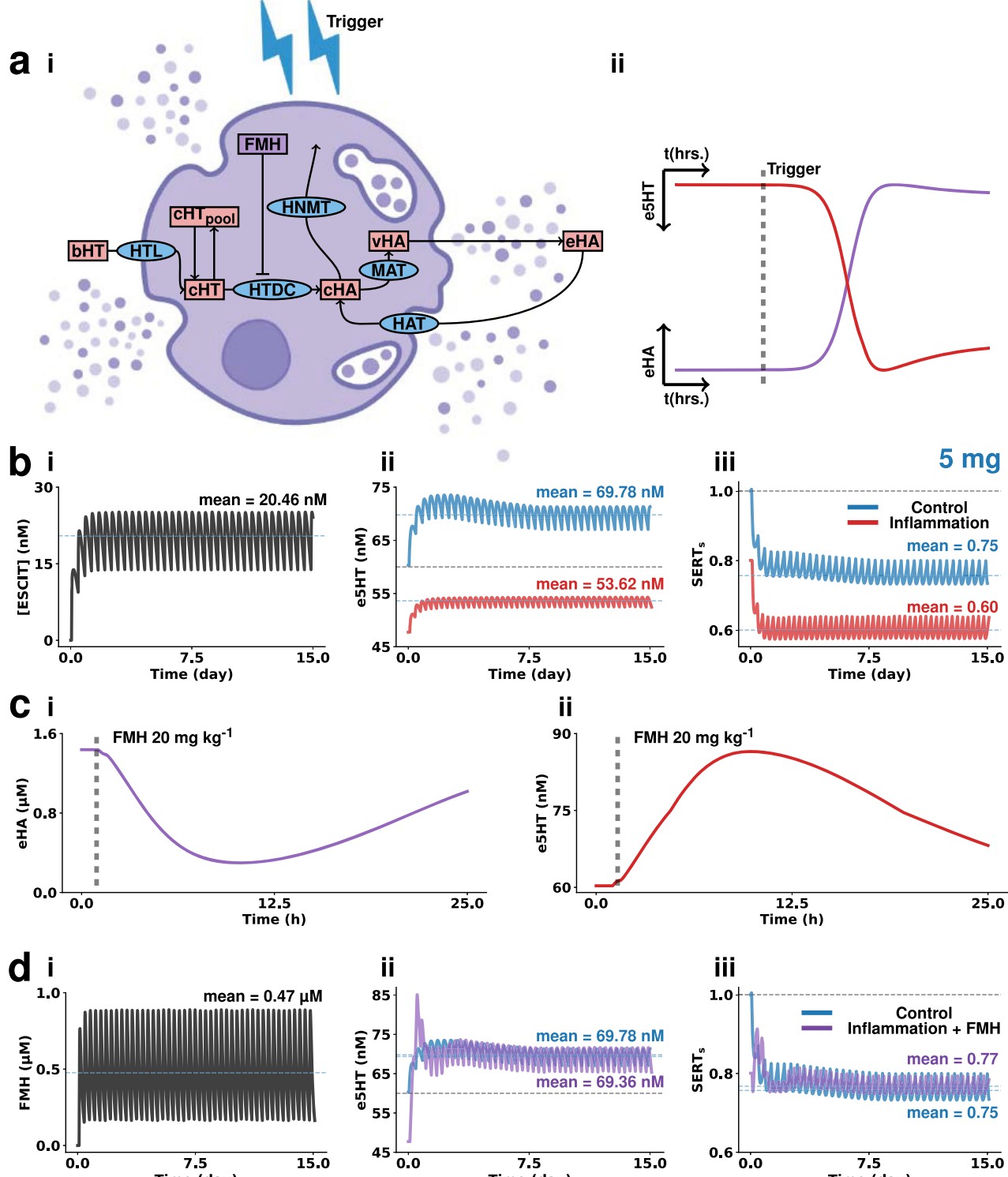

**Fig. 4 | Inflammation effects on oral chronic escitalopram efficacy. a** Mast cell model of histamine degranulation. In (i), the schematic of the computational model is given. The main variables are depicted in red rectangular labels. The acronyms of main enzymes, transporters, and receptors are represented with blue elliptic labels. Pharmacological effects are represented in purple rectangular labels. In (ii), the modeled reaction of histamine (purple) and serotonin (red) to an inflammation trigger. **b** Modeling of oral chronic dosing escitalopram (5 mg pill, ~1.02 mg kg$^{-1}$) effects on brain escitalopram (i), serotonin (ii), and SERT surface ratio (iii) during control state and inflammation. For the inflammation simulation, mast cell and glia production and release of histamine is triggered 35 days before the first dose. Administration is then repeated every 8 h. **c** Modeling of extracellular histamine (i)

and serotonin (ii) following 20 mg kg$^{-1}$ i.p. injection of (S) α-fluoromethylhistidine 1 h after the start of the simulation. **d** Modeling of oral chronic co-administration of FMH (2.5 mg analogous dose for mice, ~0.51 mg kg$^{-1}$) and escitalopram dosing effects on brain FMH (i), serotonin (ii), and SERT surface ratio (iii) during inflammation (purple) and comparison to control administration of escitalopram as described on panel B (blue). For the inflammation simulation, mast cell and glia production and release of histamine is triggered 35 days before the first dose. Administration is then repeated every 8 h. Results for the 10 mg and 20 mg human doses can be found in the Supplementary Information (Figs. S4, S5). Illustrations made with Biorender.com.

notion, we found a very clear time component; that the escitalopram concentration oscillates between pills, and the baseline concentration slowly builds up to a new steady-state after 2 days (in mice). The serotonin concentration tracks this fluctuation in real time but taking a longer period to reach steady-state (7–8 days for 5 mg). In our model, this is because the serotonin extracellular concentration is modulated by other mechanisms such as autoreceptors that also need to find a steady-state after the dose regime starts.

Autoreceptor involvement in antidepressant action is not a new idea, first being proposed by Blier, Descarries and colleagues[52,74–76]. The autoreceptor desensitization hypothesis for the delayed onset of action states that at the beginning of SSRI treatment, serotonin autoreceptors (e.g., 5-HT$_{1A}$) inhibit the firing of serotonergic blocking an increase in extracellular serotonin. In the long term, these autoreceptors internalize (desensitize) in the presence of the SSRI, firing activity is restored, allowing for an increase in extracellular serotonin. Our experimental data (and others)[6,77–80] are not consistent with this hypothesis since we observe an instant increase in serotonin levels in several brain regions[81] after SSRIs. Thus we developed our model by highlighting the coactivity between serotonin and autoreceptors. We now propose that this phenomenon does not come as a function of autoreceptor desensitization (as in the original hypotheses), but rather because of autoreceptor regulation of serotonin synthesis, release and SERT trafficking[82–88]. Autoreceptor function depends critically on serotonin binding to the receptor, and this very binding modulates extracellular serotonin via the aforementioned mechanisms. This mutual dependence is responsible for the significant time it takes for the autoreceptors and serotonin levels to both converge to a new steady-state. In practice, in our model, one equation captures the concentration of serotonin after escitalopram while a chain of equations describes the effects of the autoreceptors, meaning that at the start of chronic dosing, the escitalopram-induced increase in extracellular serotonin temporally outrun the effects of autoreceptors on regulating serotonin. This effect results in an overshoot in the serotonin concentration. Over time, the autoreceptor regulation catches up to SERT effects, and a new steady-state is reached.

Thus, our model has put forth new hypotheses an extracellular serotonin needs significant time to reach a steady-state during after chronic dosing. We next ask if the model can predict reasons for the clinical variability of escitalopram.

There are no antidepressants that are universally clinically effective. Escitalopram is considered one of the most clinically efficacious antidepressants on the market[11,89,90]. It is difficult to reconcile highly variable clinical data, but studies report patients response to escitalopram to be only 10–20% higher than placebo[91,92], which is comparable to all other antidepressants, including psilocybin[93].

The scientific community has not agreed on an explanation for this variability, spurring recent wider speculation that the monoamine hypothesis is invalid. However, there is now indisputable clinical evidence that patients presenting with inflammation are likely to be resistant to SSRIs[94–96], a fact that shines a clear light on inflammation as a relevant mechanism to consider in the pharmacodynamics of SSRIs. Indeed, in our previous experimental work, we found that an acute dose of escitalopram was less able to increase extracellular serotonin during acute and chronic inflammation (induced via LPS and chronic stress)[6,21]. In this previous work, we found that inflammation induced histamine acted on H$_3$ heteroreceptors on serotonin neurons to reduce extracellular firing. We also found that SSRIs, including escitalopram, inhibited histamine reuptake, making an SSRI less chemically effective in high histamine concentration environments (i.e., inflammation). In line with these results, Dalvi-Garcia et al. proposed a computational model suggesting that cortisolemia may render SSRIs less effective in chronic depression[97].

Here we modeled this notion in a chronic administration model. We built a simple model of histamine release in mast cells and glia as a result of an inflammatory trigger. This histamine release decreased tonic serotonin levels to a lower steady-state (which we've seen before experimentally with acute LPS and chronic stress)[21]. In this condition, the nominal escitalopram could not restore serotonin to baseline. This was not only because serotonin levels were decreased to start with, but also because the increase following escitalopram administration was much smaller when histamine was activated. An interesting point to note is that in our model, SERT density is reduced during inflammation, which is contrary to recent findings[98,99]. In our model, which does not include the effect of inflammation on SERT function/density, this feature is because of autoreceptor feedback.

A final simulation further tested this idea even further by showing that if the increase in histamine was blocked (using a histamine synthesis inhibitor), the escitalopram could be more chemically effective on raising serotonin levels. We've shown this acutely in animals previously, and now here suggest that it could also work with chronic dosing.

In summary, we have developed a new complex computational model comprising 51 equations that include allosteric binding and SERT internalization. With this model, we explained why serotonin levels take significant time to reach a new steady-state after chronic oral dosing and offered a mechanism for potential ineffectiveness of escitalopram under inflammation. Our computational model has proven to be valuable for testing experimentally complex and sometimes inaccessible concepts.

## Methods
### Experimental methods
This work focuses on a computational model, and experimental data used to guide the mathematical modeling has already been published elsewhere[22]. For an extensive description of the experimental design, methods and ethical approval see the original publication.

Color plots are the most common way to visualize FSCV data (see Fig. 1b for representative FSCV color plot) and are extensively described elsewhere[100]. Each column of the color plot represents the Faradaic current obtained with the application of a single voltage waveform. When plotted against the voltage applied, an analyte/specific signature graph called cyclic voltammogram (CV) is obtained. The waveform is repeated at 10 Hz. A row trace of the color plot represents the current at every waveform repetition for a specific potential applied, known as current vs. time trace. When extracted at the oxidation potential of interest (0.7 V for serotonin) and converted to concentration with a calibration factor, we obtain a representation of changes in concentration over the course of the experiment.

### Mathematical methods
A mathematical model was constructed, putting together previously published models and adding new proposed physiological mechanisms. The model includes earlier models of serotonin and histamine synthesis, release, and reuptake in varicosities[43,101].

More information regarding the previous models can be found in the previous publications and Supplementary Information. The complete model consists of a system of 51 differential equations. The full depiction of the model is given in the Supplementary Information. Here, a detailed description of the new additions is given. The system of differential equations was programmed in MATLAB and solved using MATLAB ODE solvers. Time is defined in hours across the whole model. The concentration of substances is defined in μM, and mass is defined in μg.

The pharmacokinetics of escitalopram after intraperitoneal injection were modeled using a four-compartment model based on prior experimental data and pharmacokinetic modeling with mice and rats[38,39]. The compartment quantities included were the peritoneum ($Q_0$), plasma ($Q_1$), brain ($Q_2$), and periphery ($Q_3$), which differential terms modeled as

$$\frac{dQ_0}{dt} = \text{inj} - k_{01}Q_0 \tag{1}$$

$$\frac{dQ_1}{dt} = k_{01}Q_0 - (k_{10} + k_{12} + k_{13})Q_1(1 - P_B) + k_{21}Q_2(1 - P_B^{\text{brain}}) + k_{31}Q_3 \tag{2}$$

$$\frac{dQ_2}{dt} = k_{12}Q_1(1 - P_B) - k_{21}Q_2(1 - P_B^{\text{brain}}) \quad (3)$$

$$\frac{dQ_3}{dt} = k_{13}Q_1(1 - P_B) - k_{31}Q_3 \quad (4)$$

where $k_{01}$ and $k_{10}$ are the rate of diffusion from the peritoneum to plasma and secretion from plasma, $k_{12}$ and $k_{21}$ are the rate between plasma and the brain and $k_{13}$ and $k_{31}$ are the rate between plasma and the periphery, $P_B$ is the ratio of escitalopram binding to protein in plasma and $P_B^{\text{brain}}$ represents the percentage of escitalopram assumed to be bound to proteins in the brain. The injection term (inj) represents the rate of increase of concentration in the peritoneum as the injection is given (in $\mu g^{-1}$), and follows the piecewise function

$$\text{inj} = \begin{cases} 0, & t < t_{\text{start}} \\ \frac{q}{t_{\text{inj}}}, & t_{\text{start}} < t < t_{\text{start}} + t_{\text{inj}} \\ 0, & t > t_{\text{start}} + t_{\text{inj}} \end{cases} \quad (5)$$

where $t_{start}$ is the time at the start of the injection, $t_{\text{inj}}$ is the duration of the injection (set as 1 s) and $q$ is the effective quantity of the drug given (in $\mu g$), which is dependent on the dose (mg kg$^{-1}$), weight of mouse (mg) and bioavailability

$$q = \text{dose} \cdot \text{weight} \cdot \text{bioavailability} \quad (6)$$

The injection term $q/t_{\text{inj}}$ makes constant the rate of increase of escitalopram in the peritoneum, so that at time $t_{\text{start}} + t_{\text{inj}}$, the total quantity given is $q$. $P_B$ and $P_B^{\text{brain}}$ were set to 0.56 and 0.15, respectively[36]. Bioavailability of escitalopram after intraperitoneal injection has not been measured experimentally and was set in the model to 0.80. The assumption behind this value is that peritoneal bioavailability is lower than that of intravenous injection (1.00) but expected to be no lower than oral administration (0.80)[37]. Mouse weight was set to 20 g, a consensus value between the C57BL/6J female mice weight (~15 g) and male mice weight (~25 g) at 8–16 weeks of age. Escitalopram dose and volume of injection modeled were dependent on mouse weight (doses: 1, 3, 10, and 30 mg kg$^{-1}$; volume: 5 mL kg$^{-1}$). Compartment volumes of peritoneum, plasma, brain, and periphery of the mouse were set to 2, 2, 0.41, and 15 mL, respectively, based on the anatomy of C57BL/6J mice and experimental notes[102].

The simulated quantity of escitalopram in the brain was then used to model the effects of the drug on SERT density and affinity (see below). The quantity of escitalopram ($\mu g$) in the brain compartment was converted to concentration ([ESCIT] in $\mu M$) using the volume of the compartment ($V_{\text{brain}}$) and molecular weight of escitalopram ($M_r = 324.39$ g mol$^{-1}$) as follows

$$[I] = 1000 \cdot \frac{Q_2}{V_{\text{brain}} \cdot M_r} \quad (7)$$

Pharmacokinetics of (S) $\alpha$-FMH were modeled using an analogous four-compartment model as the one given for escitalopram (see above). The injection function and equation used to calculate the total quantity of FMH given were also analogous. The bioavailability of FMH after intraperitoneal injection was set in the model to 0.95. Plasma protein binding ($P_B$) and brain protein binding ($P_B^{\text{brain}}$) were set to 0.60 and 0.15, respectively. To our knowledge and in contrast with escitalopram, quantitative pharmacokinetic studies of $\alpha$-FMH are limited[103–106]. As a consequence, rate constants of transport between body compartments were estimated manually based on experimental data. We iteratively changed rate constants to match our simulated concentrations in compartments to those reported in studies from[103], which contains recordings of dose-dependent plasma and brain concentration of FMH in rats and[104], which contains dose-dependent

estimations of histamine synthesis inhibition in cell cultures. Rate constants are tabulated in the Supplementary Information.

The simulated quantity of FMH was converted to concentration using the same formula as per escitalopram (see above) and the molecular weight of FMH ($M_r = 187.17$ g mol$^{-1}$).

The original pharmacokinetic model of escitalopram was built, considering experimental data from anesthetized mice after an intraperitoneal injection. Clinically, escitalopram is taken orally with a starting dose of 5–10 mg a day and can increase up to a maximum dose of 20 mg a day, independent of body weight. The dose is expected to change based on the remission of depression symptoms and side effects[107]. The clinical time regime of the dosage is designed to build a steady-state concentration of escitalopram, since the interval between dosage (24 h) is lower than the half-life of escitalopram in the plasma after oral administration[36]. We translated this clinical regime to our model based on animal experiments.

It is challenging to translate human doses to smaller animal doses. Larger animals and humans commonly have lower metabolic rates and lower doses are required to have analogous effects[108]. Because of this, it is commonly understood that conversion of a dose simply by considering changes in body weight is not a good approximation and is commonly unacceptable. Here, we used the recommended FDA guidance for the conversion of doses between animal and human trials using calculated correction factors for each species[109]. For the conversion of doses between humans and mice, we used the following equation

$$\text{Dose}_{\text{mouse}} = \text{Dose}_{\text{human}} \cdot \frac{K_m^{\text{human}}}{K_m^{\text{mouse}}} \quad (8)$$

where $K_m^{\text{human}}$ is the correction factor for humans (calculated as 37), and $K_m^{\text{mouse}}$ is the correction factor for mice (calculated as 3). $\text{Dose}_{\text{human}}$ was calculated to be 0.08, 0.17, and 0.33 mg kg$^{-1}$ by using a daily dose of 5, 10, and 20 mg, respectively, and the average global human weight of 60 kg[110]. Using the equation, $\text{Dose}_{\text{mouse}}$ was calculated to be 1.02, 2.06, and 4.11 mg kg$^{-1}$, respectively. Additionally, since the dose is modeled to be administered orally, we made modifications to the escitalopram pharmacokinetics model. The peritoneal compartment was changed to the digestive tract compartment. The rate of transport between the digestive compartment and plasma ($k_{01}$) was decreased to 0.24 h$^{-1}$, (40% of transport rate between peritoneum and plasma) to reflect the slower adsorption rate of oral administration.

To ensure a prolonged accumulation of escitalopram that mirrors the dynamics of human chronic dosing, we tailored the timing between administrations to align with clinical practices and accommodate the inherent metabolic differences between the two species. To do this, we used the drug accumulation index (RAC)[111] given by

$$\text{RAC} = \frac{1}{1 - e^{-\frac{\ln(2) \cdot t_{\text{interval}}}{t_{\frac{1}{2}}}}} \quad (9)$$

where $t_{\text{interval}}$ is the time interval between administrations, and $t_{1/2}$ is the half-life of the drug. This parameter serves as a quantification of the drug's accumulation during chronic dosing.

For the clinical chronic regime of escitalopram ($t_{\text{interval}} = 24$ h and $t_{1/2} \approx 30$ h), the resulting RAC value is ~2.35. In our simulations for mice, we opted for a $t_{\text{interval}} = 8$ h with $t_{1/2} \approx 9.69$ h, yielding a RAC value of ~2.30, which closely mimics the accumulation kinetics of clinical human dosing. The steady-state concentration ($C_{ss}$) can then be approximated by the peak concentration of a single administration ($C_{\text{single}}$), given by

$$C_{ss} \approx \text{RAC} \cdot C_{\text{single}} \quad (10)$$

Regarding FMH oral dosing, there are no standard treatments described in the literature, since it has never been used clinically. In previous

experiments, we administered twice the dose of that of escitalopram to experimentally obtain a rapid acute effect of FMH in serotonin extracellular levels[21]. Previous long-term experiments showed that at smaller chronic doses, FMH have accumulative effects on histamine synthesis[112,113]. In the model, we used an equivalent human dosing of 2.5 mg ($\text{Dose}_{\text{mouse}} = 0.51$ mg kg$^{-1}$) given every 8 h and modified the pharmacokinetics model identically to that of escitalopram to reflect oral administration (40% of $k_{01}$). The bioavailability of oral dosing was set to 0.80. A full description of the pharmacokinetic model of escitalopram and FMH can be found in Supplementary Notes 1 and 2.

Regarding serotonin and histamine co-modulation, we modeled them via a simplified mathematical depiction of G-coupled protein autoreceptors (5-HT$_{1B}$ and H$_3$ for serotonin and histamine, respectively) and activation of G-couple proteins which control the rate of synthesis and release intracellularly. The full description of the receptor signaling cascade modeling can be found in the original publications[43,101], and Supplementary Note 3, but a brief description is given here. In the model, $B_{5-HT}$ and $B_{HA}$ represent the serotonin and histamine bound to autoreceptors. $G_{5HT}$ and $G_{HA}$ represent the inactive G-protein subunit, while $G_{5HT}^*$ and $G_{HA}^*$ represent the active protein subunit, product of a serotonin or histamine molecule binding to their respective receptor. Additionally, G-protein activity is limited by RGS molecules which promote G-proteins to bind to the receptor complex[43]. $T_{5HT}$ and $T_{HA}$ represent the inactive RGS protein for serotonin and histamine receptors, respectively, while $T_{5HT}^*$ and $T_{HA}^*$ represent the active RGS protein. An increase in bound molecules stimulates the conversion of $G$ to $G^*$. At the same time, $G^*$ stimulates the conversion of $T$ to $T^*$, and $T^*$ stimulates the deactivation of $G^*$. As $G^*$ increases above a equilibrium level ($G_0^*$), a factor is calculated and multiplied to the synthesis and firing rate of serotonin and histamine neurons. For the serotonin neuron, the term for inhibition of synthesis is

$$\text{inhib}_{5HTto5HT}^{syn} = 1 - 0.1 \cdot \left( G_{5HT}^* - G_{5HT,0}^* \right) \quad (11)$$

while the inhibition of release is

$$\text{inhib}_{5HTto5HT}^{R} = 1 - 1.5 \cdot \left( G_{5HT}^* - G_{5HT,0}^* \right) \quad (12)$$

For the histamine neuron, the term of inhibition of synthesis is

$$\text{inhib}_{HAtoHA}^{syn} = 1 - 0.1 \cdot \left( G_{HA}^* - G_{HA,0}^* \right) \quad (13)$$

while the inhibition of release is

$$\text{inhib}_{HAtoHA}^{R} = 1 - 2 \cdot \left( G_{HA}^* - G_{HA,0}^* \right) \quad (14)$$

Additionally, in earlier work we hypothesized mathematically that serotonin and histamine modulate each other's release via pre-synaptic heteroreceptors based on experimental data[6,43]. Consequently, we included terms for the activation of histamine by serotonin, and the inhibition of serotonin by histamine. $B'_{5HT}$ and $B'_{HA}$ represent the serotonin and histamine bound to heteroreceptors in opposed terminals. $G'_{5HT}$ and $G'_{HA}$ represent the inactive G-protein subunit, while $G'^*_{5HT}$ and $G'^*_{HA}$ represent the active protein subunit. For the serotonin neuron, the term for inhibition of release produced by histamine binding to heteroreceptors is

$$\text{inhib}_{HAto5HT}^{R} = 1 - 3 \cdot \left( G'^*_{HA} - G'^*_{HA,0} \right) \quad (15)$$

while for the histamine neuron, the term for activation of release produced by serotonin binding to heteroreceptors is

$$\text{activ}_{5HTtoHA}^{R} = 1 + 3 \cdot \left( G'^*_{5HT} - G'^*_{5HT,0} \right) \quad (16)$$

The full mathematical description of the co-modulation mechanisms and location of the factors in the equations is given in the Supplementary Notes 3 and 4.

Regarding escitalopram pharmacodynamics, conventional orthosteric inhibition of SERTs was modeled using Michaelis–Menten enzyme kinetics. Escitalopram concentration in the brain is estimated using the pharmacokinetic model described above. Reversible and competitive inhibition in the presence of escitalopram was modeled by changing the Michaelis–Menten constant ($K_M$) of the serotonin transporter for an apparent constant ($K_M^{app}$) dependent of the concentration of the competitive inhibitor escitalopram as

$$K_M^{app} = K_M \left( 1 + \frac{[\text{ESCIT}]}{K_i} \right) \quad (17)$$

where $K_i$ is escitalopram dissociation constant (see below). Additionally, escitalopram is known to be able to bind allosterically (non-competitively) to serotonin transporters and decrease the dissociation rate on the orthosteric site, although the mechanism is not clearly defined[23]. To model this, we made $K_i$ to decay exponentially as the concentration of escitalopram increases in the brain following the expression

$$K_i = K_{i0} \cdot e^{-4 \cdot [\text{ESCIT}]} + K_{i,min} \quad (18)$$

where $K_{i0}$ is the initial escitalopram dissociation constant, set as 0.05 μM according to in vivo studies[114], and $K_{i,min}$ is the minimum dissociation constant, for which we use the measured in vitro value[115].

SERT surface density effects due to autoreceptor activation were modeled using a bidirectional transport between the plasmatic membrane and the vesicular pool of transporters, as previously described[49,82]. SERT density in each location was modeled using a ratio with respect to an equilibrium state. This is because SERT density is commonly reported in relative units[49,50,114], and there is no clear knowledge of SERT membrane concentration in serotonin varicosities. Here, we model three different locations, SERTs in the surface (SERT$_s$), SERTs in the vesicular pool (SERT$_p$) and SERTs in an inactive state (SERT$_i$). Additionally, and as reported in the literature[116,117], we model a subsection of membrane SERTs that are phosphorylated (SERT$_s^{pho}$) and available for endocytosis but maintain their reuptake function while on the membrane[118,119].

The speed of transport from membrane to pool and vice versa is dependent on the levels of g-coupled protein activated via serotonin binding to autoreceptors. As activated g-coupled protein ($G_{5HT}^*$) increases above an equilibrium level ($G_{5HT,0}^*$), the trafficking rate from pool to membrane also increases, while the inverse rate decreases. The relationship between activated g-coupled protein and trafficking rate from the surface to the pool ($k_{sp}$) and vice versa ($k_{ps}$) were calculated as

$$k_{sp} = 10 - 7.5 \cdot \left( G_{5HT}^* - G_{5HT,0}^* \right) \quad (19)$$

$$k_{ps} = 10 + 7.5 \cdot \left( G_{5HT}^* - G_{5HT,0}^* \right) \quad (20)$$

and limited to the interval between 0 and 20.

Likewise, escitalopram effects on SERT density in the membrane are modeled by a different co-located bidirectional transport between the membrane and an inactive state of the transporters. The extracellular levels of escitalopram increase the rate of trafficking between the membrane and

the inactive state ($k_{si}$) following the expression

$$k_{si} = 18.75 \cdot [\text{ESCIT}] \tag{21}$$

The effects of escitalopram on serotonin transporters are modeled as reversible, so that SERTs in the inactive state move progressively to their active state at a constant rate $k_{is} = 0.75$. These two mechanisms dictate the dynamics of SERT density in the different locations which differential terms are

$$\frac{d\text{SERT}_s^{\text{pho}}}{dt} = k_{ps}\left(G_{5HT}^*, G_{5HT,0}^*\right) \cdot \text{SERT}_p - k_{sp}\left(G_{5HT}^*, G_{5HT,0}^*\right) \cdot \text{SERT}_s^{\text{pho}} \tag{22}$$

$$\frac{d\text{SERT}_s}{dt} = \frac{d\text{SERT}_s^{\text{pho}}}{dt} + k_{is} \cdot \text{SERT}_i - k_{si}([\text{ESCIT}]) \cdot \text{SERT}_s \tag{23}$$

$$\frac{d\text{SERT}_p}{dt} = k_{sp}\left(G_{5HT}^*, G_{5HT,0}^*\right) \cdot \text{SERT}_p^{\text{pho}} - k_{ps}\left(G_{5HT}^*, G_{5HT,0}^*\right) \cdot \text{SERT}_p \tag{24}$$

$$\frac{d\text{SERT}_i}{dt} = k_{si}([\text{ESCIT}]) \cdot \text{SERT}_s - k_{is} \cdot \text{SERT}_i \tag{25}$$

Finally, the rate of reuptake of SERTs is then calculated based on the density of transporters ($\text{SERT}_s$) in the membrane and the apparent affinity ($K_M^{\text{app}}$)

$$V_{\text{SERT}} = \frac{250 \cdot e5HT}{K_M^{\text{app}} + e5HT} \cdot \text{SERT}_s \tag{26}$$

In the present model, both the histamine and serotonin varicosities possess a readily releasable pool (RRP) of vesicles that are secreted during firing and a reserve of vesicles which replenishes the RRP when its diminished, as described theoretically[120]. This was modeled by a unidirectional transport of vesicles from the reserve pool to the RRP when the RRP concentration levels change from an expected of equilibrium value. The rate of trafficking between the serotonin reserve and the RRP was given by

$$V_{\text{traff}}^{5HT} = 15 \cdot \left(v5HT_0 - v5HT\right) \tag{27}$$

While for histamine, this rate was given by

$$V_{\text{traff}}^{HA} = 15 \cdot \left(vHA_0 - vHA\right) \tag{28}$$

and both were limited to the interval between 0 (no trafficking) and 100 (maximum trafficking). Additionally, the model was changed so that both the reserve pool and RRP are reloaded with neurotransmitter molecules from the cytosol by the monoamine transporter (see Supplementary Notes 3 and 4).

Regarding neuroinflammatory processes, glial cells were modeled to be able to synthesize and passively release histamine (via membrane leaking), in addition to being able to reuptake and metabolize histamine, included previously[101]. As for the histamine neuron, we included in glia all the mechanisms required to synthesize histamine; a histidine transporter, which carries histidine from the bloodstream to the cytosol of glia, and histidine carboxylase, which converts cytosolic histidine into histamine. In all cases, the affinity of the enzymes was kept equal to the same enzyme in histamine neurons, while the capacity (which depends on the quantity of enzymes) was modified to fit experimental data.

The histidine transporter in glia was set to have half the capacity ($V_{\text{max}} = 2340\,\mu\text{M h}^{-1}$) and the same affinity ($K_M = 1000\mu M$) of that in histamine neurons (see Supplementary Note 4), so that the speed of

histidine transport follows the Michaelis–Menten rate of reaction

$$V_{\text{HTL}}^g = \frac{2340 \cdot b\text{HT}}{b\text{HT} + 1000} \tag{29}$$

where $V_{\text{HTL}}^g$ is the speed of histidine transport from blood histidine ($b$HT) to glia cytosolic histidine ($g$HT) in micromolar per hour. Histidine decarboxylase in glia was modeled with $V_{\text{max}} = 61.42\,\mu\text{M h}^{-1}$ and $K_M = 270\mu M$, so that the speed of production of histamine in glia was

$$V_{\text{HTDC}}^g = \frac{61.42 \cdot g\text{HT}}{g\text{HT} + 270} \tag{30}$$

Additionally, glia was modeled to also have a histidine pool just as programmed for histamine neurons (see Supplementary Note 4).

Mast cell dynamics were modeled in a similar manner to glia, although distinct based on knowledge of these two types of cells. Mast cells were modeled to be able to obtain histidine from the bloodstream and synthesize and metabolize histamine. The histidine transporter speed of histidine transport to the mast cell cytosol is

$$V_{\text{HTL}}^{\text{MC}} = \frac{109.5 \cdot b\text{HT}}{b\text{HT} + 1000} \tag{31}$$

Histidine decarboxylase capacity was set as $V_{\text{max}} = 877.50\,\mu\text{M h}^{-1}$ and affinity $K_M = 270\mu M$, so the speed of production of histamine in mast cells was

$$V_{\text{HTDC}}^{\text{MC}} = \frac{877.50 \cdot c\text{HT}^{\text{MC}}}{c\text{HT}^{\text{MC}} + 270} \tag{32}$$

Histamine methyltransferase in mast cells was modeled with capacity $V_{\text{max}} = 21.20\,\mu\text{M h}^{-1}$ and affinity $K_M = 4.2\mu M$, so that the rate of histamine metabolism was

$$V_{\text{HNMT}}^{\text{MC}} = \frac{21.20 \cdot c\text{HA}^{\text{MC}}}{c\text{HA}^{\text{MC}} + 4.2} \tag{33}$$

where $c\text{HA}_{mc}$ represents the cytosolic concentration of histamine in mast cells. Additionally, mast cells were modeled to reuptake histamine from the extracellular space via a histamine putative transporter with capacity $V_{\text{max}} = 3375\,\mu\text{M h}^{-1}$ and affinity $K_M = 10\mu M$, so that the rate of reuptake was

$$V_{\text{HAT}}^{\text{MC}} = \frac{3375 \cdot e\text{HA}}{e\text{HA} + 10} \tag{34}$$

where $e$HA represents the extracellular concentration of histamine. Finally, the vesicular monoamine transporter traffic from intracellular histamine to vesicular histamine was modeled with a $V_{\text{max}} = 21104\,\mu\text{M h}^{-1}$ and $K_M = 24\mu M$, as well as a linear back-leak from vesicles to intracellular histamine

$$V_{\text{MAT,HA}}^{\text{MC}} = \frac{21104 \cdot c\text{HA}^{\text{MC}}}{c\text{HA}^{\text{MC}} + 24} - 5 \cdot c\text{HA}^{\text{MC}} \tag{35}$$

Unlike glia, mast cells release of histamine was programmed to be residual during control simulations (degranulation at 0.01% its maximum capacity) and activate progressively following the sigmoid function

$$k_{\text{inf}} = \frac{1}{1 + e^{-20 \cdot (t - t_h)}} \tag{36}$$

with $t_h$ being the time at which the function is at 50% of its maximum capacity, given by

$$t_h = t_{start} + \frac{\ln(999)}{20} \tag{37}$$

where $t_{start}$ is the time at which the inflammation activity of mast cells starts increasing from 0.01%. Additionally, the synthesis of histamine by mast cells and glia were also enhanced following the same sigmoid function during neuroinflammation.

Regarding FMH pharmacodynamics, FMH is an irreversible inhibitor of histidine decarboxylase, the main enzyme responsible for the synthesis of histamine. The mechanism of inactivation of the enzyme has been previously proposed[121], and it is explained in detail in the original publication. Here, just a general description of the process is given. Briefly, FMH can inhibit histidine decarboxylase first by binding to the enzyme ($k_1$). This step results in the decarboxylation of FMH. After that, FMH can lose a fluoride ion following a nucleophilic attack, forming a highly reactive compound that binds covalently to the enzyme ($k_2$). A series of other hypothesized processes (e.g., transamination) could lead to a non-reactive product, in which case the enzyme would not be inactivated ($k_3$). Assuming these three processes, the differential term of active HTDC enzyme follows

$$\frac{dHTDC_a}{dt} = -k_{FMH}([FMH]) \cdot HTDC_a + HTDC_{in}(HTDC_a) \tag{38}$$

$$k_{FMH} = k_1 \cdot \frac{k_2}{k_2 + k_3} \cdot \frac{[FMH]}{K_i + [FMH]} \tag{39}$$

where $HTDC_a$ is the ratio of active HTDC enzyme, [FMH] is the concentration of FMH assumed from the pharmacokinetics model, $K_i$ is the dissociation constant of the FMH-HTDC complex (set to 8.3 μM) and $k_{FMH}$ is the inactivation rate, derived the rate constants of the processes, the dissociation constant and concentration of FMH. $HTDC_{in}$ is the rate of replenishment of enzyme via synthesis of the protein. This replenishment follows the function

$$HTDC_{in} = 0.55 \cdot (1 - HTDC_a) \tag{40}$$

The estimated ratio of active HTDC enzyme is then multiplied to the velocity of the enzyme in histamine neurons ($V_{HTDC}$), glia ($V_{HTDC}^g$), and mast cells ($V_{HTDC}^{MC}$) (see equations in Supplementary Notes 4 and 5).

### Statistics and reproducibility
Given that our work presented in the paper involves simulations of a mathematical model rather than empirical data, it's important to note that statistical comparisons were not applied in this context.

Regarding the analysis of data, to compare simulations to FSCV experimental signals, simulation of the model responses to firing stimulation over 30 s were subtracted to have a zero baseline. This was achieved by subtracting the average of the first 5 s (prior to the simulation of electrical stimulation) to each time point in the 30 s trace.

All pharmacokinetic parameters were calculated using Python 3.11 and the NumPy and SciPy libraries. Maximum concentration ($C_{max}$) of escitalopram and extracellular serotonin in the brain were calculated, extracting the maximum value from each simulation. Area under the curve ($AUC_{25h}$) of concentration of escitalopram and serotonin over time was calculated using the Simpson's rule and used as an estimation of overall drug exposure. The area was calculated over 25 h of simulation. The half-life ($t_{1/2}$) was calculated fitting an exponential decay function from $C_{max}$ to the end of the simulated concentration trace following

$$C(t) = C_0 e^{-\lambda t} \tag{41}$$

Where $t_{1/2}$ was calculated from the decay rate constant ($\lambda$) as

$$t_{1/2} = \frac{\ln(2)}{\lambda} \tag{42}$$

The fitting was performed using the Levenberg–Marquardt algorithm. Regression lines were obtained using the linear least squares method. Maximum amplitude ($Amp_{Max}$) and half-life of evoked traces was calculated following analogous methodology in a custom-designed application[122].

### Reporting summary
Further information on research design is available in the Nature Portfolio Reporting Summary linked to this article.

### Data availability
Due to the size of files, simulation data and other files are available upon request. A subsection of the files can be found in zenodo (https://doi.org/10.5281/zenodo.8406456)[123]. Additionally, available code (see below) can be used to generate the data used in the study.

### Code availability
Python files for the processing of selected signals, parametric analysis and plotting is available in zenodo (https://doi.org/10.5281/zenodo.8406456)[123]. Additionally, a full description of the system of differential equations used to model the signals is given in the Supplementary Information, and the code used is provided in the zenodo repository above. Mathematical simulations were run on MATLAB version 2023b. Data processing and figures were obtained using Python version 3.11.

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

## Acknowledgements

We thank Dr. Melinda Hersey for the acquisition of the experimental FSCV signals used to guide the fitting of our model. This study was funded by the National Institutes of Health R01MH106563 (P.H.) and the CAMS Lectureship Award (P.H.).

## Author contributions

P.H. and S.M. conceptualized the work, wrote the manuscript, and generated the figures. A.C. and M.R. developed the model of histamine and serotonin receptor co-modulation. J.B., M.C.R., and N.H.F. developed the original models of serotonin and histamine synthesis, release and reuptake, and aided with the implementation of this version of the model. S.M. developed the PK/PD model of escitalopram and FMH and linked the different models together. The manuscript was proofread by all the authors.

## Competing interests

The authors declare no competing interests.
