## [Peer Review File · Communications Biology]

Reviewers' comments:

Reviewer #1 (Remarks to the Author):

Summary:

This manuscript presents a theoretical model of the effects of escitalopram, an SSRI, on serotonin transporter expression under inflammatory conditions. The model is based on a comprehensive understanding of the synthesis, release, and reuptake of serotonin, as well as the effects of histamine and other factors on these processes. The authors use the model to simulate chronic oral escitalopram in mice and explore the complexities involved in replicating human disease and clinical dosing in animal models. The paper sheds light on the challenges of developing effective SSRIs and highlights the need for further research in this area.

Impression:

This is a highly relevant work at this time, given the recent evidence linking psychiatric disorders, especially major depression, and inflammation. The group has been consistently developing well-described mathematical models in this area for some time, and this work is no exception. All equations and assumptions are thoroughly explained and justified. I would like to offer a few considerations to contribute to the text.

- 1) In the second, third, and fourth paragraphs of the Results section (lines 92, 101, and 129, respectively), there is a description of the steps taken to reach the satisfactory model proposed in the article. While this description is important, it seems a little bit lengthy and often followed by "but it didn't work.". I suggest that these passages be rewritten in a more concise manner.
- 2) In the caption of Figure 3, subfigure C is not referenced. Instead, subfigure B is referenced twice.
- 3) In several parts of the text, even when discussing information presented in tables, there are references to the colors of variables in the figures. I suggest that this color information be omitted from the main text and be included only in the captions of the figures.
- 4) On line 266, I don't see how Figure 4Bi is analogous to Figure 3C while simultaneously depicting chronic escitalopram performance under inflammation. I would suggest changing it to 'Figure 4B is analogous...', without the 'i'.
- 5) While the unit mgkg^{-1} can be easily understood by one familiar with this notation, I would recommend changing it to $\text{mg}\cdot\text{kg}^{-1}$ or mg kg^{-1} (as in other parts of the paper) or mg/kg . This is only a suggestion, and I leave this decision to the author's discretion.
- 6) There is a missing citation at line 437.
- 7) On line 459: "X differential equations..."
- 8) On line 498, instead of 'input,' did the authors mean 'inj'?
- 9) What does "supra" refer to on lines 535 and 548?
- 10) I believe that, in terms of mathematical models for depression, the article 'A model of dopamine and serotonin-kynurenine metabolism in cortisolemia: Implications for depression' (Dalvi-Garcia, F et al., 2021) deserves citation. While the discussed mechanisms differ from the present study, Dalvi-Garcia et al. (2021) observed through their model the ineffectiveness of SSRIs in major depression under cortisolemia and a link with inflammation.

Reviewer #2 (Remarks to the Author):

Overall, the manuscript presents a very interesting topic with relevant insights to the general pharmacology community. The authors have successfully developed a computational model of escitalopram's PK and PD. More importantly, the authors have demonstrated the utility of in-silico model in knowledge integration and hypothesis testing. However, what worries me is that the readers with general pharmacology background may be struggling with some context presented in the manuscript. The authors still need to improve the readability of their work. Moving math into supplementary material and describing the model in a plain language could be an option. My critiques are outlined below:

1. The current manuscript could be difficult for general pharmacologists to understand. I suggest you to keep less math in the main article.
2. Please correct me if I have missed something. I didn't find the model validation part for the chronic model of escitalopram's PK and PD. Everything seems to be model simulation. Since there is a big gap between acute and chronic escitalopram administration, it would be great if the authors could design some experiments based on model simulation and compare the wet-lab observation with in-silico results. This would largely increase readers' confidence in the developed model.

Answer to Reviewer #1 Comments

- 1) In the second, third, and fourth paragraphs of the Results section (lines 92, 101, and 129, respectively), there is a description of the steps taken to reach the satisfactory model proposed in the article. While this description is important, it seems a little bit lengthy and often followed by "but it didn't work.". I suggest that these passages be rewritten in a more concise manner.

Thank you for this comment, and while we would ordinarily agree, if this is not a big sticking point for the reviewer, we would like to keep it this way. We had to perform 3 major iterations of the model, and the 2 times it didn't work well we learned important physiological things that we feel should be shared explicitly.

- 2) In the caption of Figure 3, subfigure C is not referenced. Instead, subfigure B is referenced twice.

Thank you for spotting this. We have fixed this typo.

- 3) In several parts of the text, even when discussing information presented in tables, there are references to the colors of variables in the figures. I suggest that this color information be omitted from the main text and be included only in the captions of the figures.

Thank you for your suggestion. We have removed the references to the colours of the variables outside of figure captions.

- 4) On line 266, I don't see how Figure 4Bi is analogous to Figure 3C while simultaneously depicting chronic escitalopram performance under inflammation. I would suggest changing it to 'Figure 4B is analogous...', without the 'i'.

Thank you for spotting this. We have made the suggested changes in the text.

- 5) While the unit mgkg⁻¹ can be easily understood by one familiar with this notation, I would recommend changing it to mg.kg⁻¹ or mg kg⁻¹ (as in other parts of the paper) or mg/kg. This is only a suggestion, and I leave this decision to the author's discretion.

Thank you for spotting this. We have standardised this to mg kg⁻¹ all across the manuscript text and supporting information. Additionally, we have also changed it on Figure 1, Figure 4 and Figure S1.

- 6) There is a missing citation at line 437.

Thank you for spotting this. We have added the citation.

7) On line 459: “X differential equations...”

Thank you for catching this. We have fixed this typo in the text.

8) On line 498, instead of 'input,' did the authors mean 'inj'?

Correct, thank you for seeing this. We have fixed in the manuscript.

9) What does "supra" refer to on lines 535 and 548?

Vide supra means “see above” and in both cases it is used to refer to the escitalopram pharmacokinetics described before the FMH pharmacokinetics. We have changed vide supra for see above in the text.

10) I believe that, in terms of mathematical models for depression, the article 'A model of dopamine and serotonin-kynurenine metabolism in cortisolemia: Implications for depression' (Dalvi-Garcia, F et al., 2021) deserves citation. While the discussed mechanisms differ from the present study, Dalvi-Garcia et al. (2021) observed through their model the ineffectiveness of SSRIs in major depression under cortisolemia and a link with inflammation.

Thank you for your suggestion. We agree that the work by Dalvi-Garcia and colleagues deserves a citation. We have included a little discussion under the subsection “Under Inflammation, Therapeutic Doses of Escitalopram Cannot Restore Serotonin Levels”:

“In line with these results, Dalvi-Garcia et al. proposed a computational model suggesting that cortisolemia may render SSRIs less effective in chronic depression⁹⁷.

Answer to Reviewer #2 Comments

1. The current manuscript could be difficult for general pharmacologists to understand. I suggest you keep less math in the main article.

We completely agree with this notion and spent many hours taking the math out of the results and discussion sections. We do need to have the maths though in the methods, because the entire work is computational. To make it easier for the readers to navigate we split the methods into maths vs non maths methods.

2. Please correct me if I have missed something. I didn't find the model validation part

for the chronic model of escitalopram's PK and PD. Everything seems to be model simulation. Since there is a big gap between acute and chronic escitalopram administration, it would be great if the authors could design some experiments based on model simulation and compare the wet-lab observation with in-silico results. This would largely increase readers' confidence in the developed model.

This is a really important key part of our work. In reality it is impossible to experimentally model the chronic regime in mice, as we wrote in the paper "Animals will not willingly swallow a pill and other experimental ways to administer a chronic dose fall short in reproducing the pharmacokinetics of a human dose regime. This is because in humans, the concentration of the active agent oscillates around a period between a single pill administration, dependent on the pharmacokinetics of the agent and the characteristics of agent release from the pills." This reason drove the motivation for the whole work (i.e. to model something that can't be captured experimentally).

REVIEWERS' COMMENTS:

Reviewer #1 (Remarks to the Author):

Although there is an complex mathematical framework, it is part of the used modeling technique and the results are of great relevance to the field of mental health and pharmacology. The authors made the suggested corrections and addressed the main suggestions indicated in the review, so this work deserves to be published.

Reviewer #2 (Remarks to the Author):

My previous comments have been addressed. Thanks for your hard work!